# Oncostatin M: From Intracellular Signaling to Therapeutic Targets in Liver Cancer

**DOI:** 10.3390/cancers14174211

**Published:** 2022-08-30

**Authors:** Alessandra Caligiuri, Stefano Gitto, Giulia Lori, Fabio Marra, Maurizio Parola, Stefania Cannito, Alessandra Gentilini

**Affiliations:** 1Department of Experimental and Clinical Medicine, University of Florence, 50139 Florence, Italy; 2Department of Clinical and Biological Sciences, Unit of Experimental Medicine & Clinical Pathology, University of Torino, 10125 Torino, Italy

**Keywords:** cholangiocarcinoma, hepatocellular carcinoma, signaling, tumor microenvironment, tumor proliferation, epithelial–mesenchymal transition, inflammation

## Abstract

**Simple Summary:**

Pro-inflammatory cytokines play a key role in innate- and adaptive-immunity-mediated liver carcinogenesis. Among these, oncostatin M (OSM) critically contributes to physiological and pathological processes, including extracellular matrix remodeling, hematopoiesis, differentiation, inflammatory response, proliferation, acquisition of cancer stem cell markers, drug resistance, and metastatic phenotype. Here, we review the current knowledge on the role of OSM in liver cancers, focusing on recent progress in the understanding of the molecular mechanisms of IL-6-type cytokine signaling cascades.

**Abstract:**

Primary liver cancers represent the third-most-common cause of cancer-related mortality worldwide, with an incidence of 80–90% for hepatocellular carcinoma (HCC) and 10–15% for cholangiocarcinoma (CCA), and an increasing morbidity and mortality rate. Although HCC and CCA originate from independent cell populations (hepatocytes and biliary epithelial cells, respectively), they develop in chronically inflamed livers. Evidence obtained in the last decade has revealed a role for cytokines of the IL-6 family in the development of primary liver cancers. These cytokines operate through the receptor subunit gp130 and the downstream Janus kinase/signal transducer and activator of transcription (JAK/STAT) signaling pathways. Oncostatin M (OSM), a member of the IL-6 family, plays a significant role in inflammation, autoimmunity, and cancer, including liver tumors. Although, in recent years, therapeutic approaches for the treatment of HCC and CCA have been implemented, limited treatment options with marginal clinical benefits are available. We discuss how OSM-related pathways can be selectively inhibited and therapeutically exploited for the treatment of liver malignancies.

## 1. Introduction to Liver Cancer

Primary liver cancers (PLCs), the third-leading cause of tumor-related mortality worldwide, include hepatocellular carcinoma (HCC) (arising from hepatocytes, accounting for 80–90% of PLC), cholangiocarcinoma (CCA) (originating from biliary epithelial cells, accounting for 10–15% of PLC) [1], and mixed hepatocellular–cholangiocellular carcinoma (accounting for 0.4–14.2% of PLC), which shows markers of hepatocellular carcinoma and cholangiocarcinoma [2].

### 1.1. Hepatocellular Carcinoma

According to Global Cancer Incidence, Mortality and Prevalence (GLOBOCAN) 2020, HCC is the sixth-most-common cancer and the third-most-common cause of cancer-related mortality worldwide [3]. HCC has a very poor clinical outcome, often due to a delay in diagnosis (early stages being asymptomatic) and resistance to conventional chemotherapy and radiotherapy, with the majority of patients diagnosed at advanced stages and then not eligible for curative therapy [3,4,5]. HCC develops on the background of a cirrhotic liver, is intimately related to persistent fibrogenesis and inflammatory response, and may be associated with chronic viral infection by hepatitis B virus (HBV) and hepatitis C virus (HCV), non-alcoholic fatty liver disease (NAFLD), alcohol abuse, and, to a lesser extent, autoimmune disorders or exposure to aflatoxin B1 [3]. HCC incidence and mortality are expected to increase in the near future as a consequence of the close association between the obesity pandemic and NASH, which is emerging as the most rapidly rising cause of HCC and is predicted to amplify the incidence of HCC up to 41% by 2040 [3,6]. Genetic (somatic mutations and chromosomal aberrations leading to chromosomal instability) and epigenetic alterations (DNA methylation) can contribute to HCC development [7]. Mutations of the TERT promoter, AXIN1, and CTNNB1 genes have been observed in dysplastic nodules and established HCC [8]. Moreover, alterations in p53 activation and in proteins involved in cell cycle control (CDKN2A), as well as aberrant activation of the mitogen-activated protein kinase (MAPK) and phosphoinositide 3-kinase-AKT-mammalian target of rapamycin (PI3K-AKT-mTOR) pathways have been usually described in HCC of different etiologies [8]. The complex tumor microenvironment (TME), through the interactions of tumor cells, stromal cells, innate and adaptive immune cells, and the involvement of hypoxic conditions, plays a critical role in HCC development by modulating fibrogenesis, epithelial–mesenchymal transition (EMT), invasiveness, and metastasis [9]. A deep investigation of TME-related issues can disclose new therapeutic opportunities to counteract HCC development and progression.

### 1.2. Cholangiocarcinoma

CCA, the second-most-common primary liver cancer, is a very aggressive cancer acncounting for 2% of all cancer-associated deaths worldwide yearly [10,11] and includes a heterogeneous group of malignancies occurring at any point of the biliary tree. CCAs may derive from cholangiocytes of biliary tracts, peribiliary glands, and, likely, progenitor cells or even hepatocytes [2,11,12,13]. CCAs are classified on an anatomical basis as intrahepatic CCA (iCCA) and extrahepatic CCA (eCCA), the latter still distinguished into perihilar CCA (pCCA) and distal CCA (dCCA). iCCA is located above the second-order bile ducts, then originating from either segmental bile ducts or smaller branches of the intrahepatic biliary tree. pCCA, also known as a Klatskin tumor, may arise in the right and/or left hepatic duct and/or at their junction, while dCCA arises from the common bile duct [11,14]. CCA is usually asymptomatic in the early stages (in particular iCCA), is diagnosed at an advanced phase when curative options are lacking, is resistant to common chemotherapy, and has a very poor prognosis [11]. Potentially curative surgical treatment is limited to the small group of patients with early-stage disease (approximately 35%) [12]. CCA incidence is high in the Far East (100 per 100,000 people for men and 50 per 100,000 people for women), but much lower in Western countries (1–2 cases per 100,000) due to differences in risk factors and association with genetic, ethnic, and environmental issues [15,16]. Moreover, epidemiological reports have outlined an increase (up to ten-fold) in the iCCA global occurrence and mortality, whereas the incidence of eCCA remained unchanged or slightly decreased [17]. The pathogenesis of CCA results from multiple factors, with genetic and environmental issues playing a critical role. CCA mainly develops in a background of chronic biliary inflammation, such as primary sclerosing cholangitis (PSC) [18]. Chronic hepatitis B and C, alcohol abuse, cigarette smoke, cirrhosis, obesity, and diabetes have been also related to the development of iCCA. Moreover, several genomic and epigenetic alterations characterize CCA development, although the interpretation of the data is difficult due to ethnic, geographical, and etiological differences or to the use of different detection techniques or misclassifications [19]. CCA development is associated with the deposition of a considerable amount of fibrous stroma, creating a TME in which the different types of infiltrating cells (cancer-associated fibroblasts—CAFs, tumor-associated macrophages—TAMs, aberrant immune cells) facilitate tumor progression and invasion of the normal parenchyma [20,21]. Indeed, the release of several soluble mediators and the expression of the cognate receptors have been shown to significantly contribute to tumor–stroma interactions [20,21,22,23,24,25]. Despite an improved understanding of CCA pathogenesis occurring recently and the parallel disclosure of new therapeutic targets, translation into clinical trials is limited due to the small subset of patients presenting druggable alterations.

## 2. Oncostatin M: ID Card

Pro-inflammatory cytokines such as tumor necrosis factor (TNF)-α and interleukin (IL)-6, as well as their signaling pathways, involving nuclear factor kappa B (NF-κB), c-Jun NH_2_-terminal kinase (JNK), and signal transducer and activator of transcription 3 (STAT3), play a key role in innate- and adaptive-immunity-mediated liver carcinogenesis [26]. In particular, IL-6 and STAT3 overexpression has been related to both HCC and CCA promotion and development, pointing out these molecules as potential therapeutic targets [26,27,28].

Oncostatin M (OSM) is a 26 kDa molecular weight multifunctional cytokine belonging to the IL-6 (or gp130) cytokine family, which includes IL-6, leukemia inhibitory factor (LIF), IL-11, IL-27, IL-30, IL-31, ciliary neurotrophic factor (CNTF), neuropoietin-1 (NP-1), and cardiotrophin 1 (CT-1). OSM, which shares high structural, functional, and genetic homology with LIF [29,30], is a unique cytokine, which critically contributes to physiological and pathological processes, including extracellular matrix remodeling, hematopoiesis, differentiation, inflammatory response, proliferation, acquisition of cancer stem cell (CSC) markers, drug resistance, and achievement of metastatic phenotype [29,30,31]. OSM, although mainly produced by activated cells of innate and adaptive immunity (monocytes/macrophages, neutrophils, dendritic cells, T lymphocytes, hematopoietic, and mesenchymal cells) [29,31,32] can be secreted also by cancer cells [33]. OSM operates through two different heterodimeric receptors: the type I receptor, formed by gp130 and the LIF receptor β (LIFRβ), and the type II receptor, formed by gp130 and the OSM receptor β (OSMRβ) (Figure 1), which are differentially engaged in human and mouse. Human OSM can bind to either gp130/OSMRβ or gp130/LIFRβ, whereas murine OSM binds only the type II receptor [34,35]. Of interest, the OSMRβ/LIFRβ ratio and their different expression on various cell types represent the mechanism by which OSM and LIF can exert common biological functions in some tissues and distinct unique actions in others [35].

OSM receptor complexes lack intrinsic kinase activity, and OSM–receptor interaction and receptor hetero-dimerization lead to Janus kinases’ (JAKs) recruitment and activation, by transphosphorylation on the receptor intracellular domain [29,31,35]. JAKs’ recruitment activates intracellular signaling pathways and stabilizes the OSM receptor complex: the binding of JAK1 to the OSMRβ is crucial for masking hydrophobic amino acids, which usually regulate the confinement of OSMRβ in the endoplasmic reticulum [29,32,36].

The C-terminal region of receptor types I and II contains tyrosine motifs, which, phosphorylated by JAK1/2, act as a docking site for STAT1 and STAT3. Moreover, STAT1 tyrosine phosphorylation can be independent of the OSMRβ receptor tyrosine motifs and due to a direct action of JAKs [29]. Moreover, OSM, unlike the IL-6 family of cytokines, but similar to IL-4 and IL-13, can activate STAT6 in a cell-type-specific way [29,37]. Furthermore, OSM can recruit, on a conserved OSMRβ-Tyr861 residue, the adapter sarcoma (Src) homology and collagen (Shc) protein, which, in turn, can activate other downstream proteins such as extracellular-regulated kinase 1/2 (ERK1/2), p38, or JNK [31,32,35,38,39]. OSM has also been reported to activate the PI3K/Akt pathway [39] and PKCδ [40] through mechanisms still poorly characterized. OSM can also negatively modulate MAP kinase cascades by recruiting the tyrosine phosphatase SHP-2 on Tyr759 and Tyr974 of the type I receptor, as well as the suppressors of cytokine signaling (SOCS), in particular SOCS3, on Tyr759, or through the direct action of JAK1/2 [31,32,35]. An overview of the signaling pathways activated by OSM is reported in Figure 2.

## 3. Biological Activity of Oncostatin M

OSM is a pleiotropic cytokine that can affect several biological processes depending on the cell type, including inflammatory response, proliferation, EMT, invasiveness, metastasis, and CSC behavior.

### 3.1. Role of OSM in Inflammation

OSM plays a major role in inflammation, being involved in the acute phase response and in chronic inflammatory conditions, leading eventually to tissue fibrosis and cancer, as well as being expressed by immune cells in response to a variety of soluble mediators [41]. OSM can act either directly recruiting/activating innate immune cells or, indirectly, by regulating stromal cells located at the injured areas. Depending on the cellular context and the intracellular signaling elicited, OSM may display either anti- or pro-inflammatory activities [42,43]. Murine and human recombinant OSM have been used in murine models to reveal OSM’s contribution to inflammation. Since human recombinant OSM can bind only the murine type I receptor and activate the related signaling pathway [44], the anti-inflammatory action of OSM operates through this receptor complex. Accordingly, OSM released by neutrophils in inflamed tissues can inhibit IL-1-induced expression of IL-8, then inhibiting local neutrophil infiltration and stimulating proliferation and collagen release by dermal fibroblasts, speeding up the wound healing process in a diabetic murine model [42,43]. Intramuscular OSM injection in mice, via adenoviral-OSM (AdOSM) delivery, resulted in IL-33’s increased secretion by liver endothelial cells and consequent expansion/activation of liver CD4+ ST2+ lymphocytes exhibiting hepato-protective role [45]. Similarly, intravenous injection of AdOSM in diethyl-nitrosamine (DEN)-treated rats protected them from liver damage, likely due to the pro-regenerative action exerted by OSM on hepatocytes [46]. In contrast, repetitive intravenous vector-delivered OSM administration in healthy mice resulted in hepatic fibrosis, whereas OSM-deficient mice were protected from thioacetamide-induced liver fibrosis [47].

OSM has been associated with pro-inflammatory activities, inducing target cells to release cytokines and chemokines such as CXCL3, CCL2, CCL5, and CCL20 [48], which further recruit neutrophils and monocytes/macrophages, thus creating positive feedback, which amplifies OSM’s effects. Accordingly, in chronic inflammatory conditions, OSM/OSMβR are often overexpressed and OSM sustains inflammation and promotes fibrosis. Notably, OSM is the only IL6-family member able to bind extracellular matrix (ECM) molecules (collagens, fibronectin, laminin), upregulating their expression and remaining localized in high concentrations at the site of injury [49,50].

In tumors, OSM and other members of the IL-6 cytokine family can exhibit indirect pro-tumorigenic effects affecting the tumor microenvironment and stromal cells and modulating inflammatory and immune responses [51]. In a murine model carrying a specific deletion of von Hippel-Lindau (VHL)-deficient renal tubular cells, OSM led to endothelial cells’ activation, favoring the recruitment and polarization of macrophages, then generating an inflammatory and tumorigenic microenvironment and promoting the development of clear-cell renal carcinoma [52]. At present, a role for the OSM/OSMR axis in reconstituting a milieu suitable to cancer progression has been reported in pancreatic ductal adenocarcinoma, with macrophage-derived OSM upregulating pro-inflammatory genes in CAFs, leading to tumor growth and metastasis [53]. In breast carcinoma, hypoxic epithelial tumor cells overexpress and release high levels of OSM, favoring the recruitment and M2 polarization of macrophages [54]. Moreover, myeloid-cell-derived OSM can drive CAFs to a pro-carcinogenic phenotype, showing increased contractility and secretion of soluble mediators, leading to immune cells’ accumulation and further supporting breast cancer development [55].

In the liver, immuno-histochemical analysis of liver specimens showed high OSM positivity in cirrhotic patients in cells of hepatic sinusoids also positive for the CD68 macrophage marker [56]. Moreover, Znoyko and colleagues observed that only the OSM receptor type I was upregulated in human specimens from cirrhotic patients, suggesting its involvement in fibrogenic progression [47]. Matsuda and colleagues reported that OSM promoted liver fibrogenesis, inducing tissue inhibitors of metalloproteinases’ (TIMPs) production and type I collagen deposition by activated hepatic stellate cells (HSCs), as well as by favoring the M2 phenotypic switch of macrophages [47]. Yang and collaborators showed that, in a rat model of DEN-induced HCC, hepatic overexpression of OSM by adenovirus resulted in increased levels of aminotransferases, suggesting that OSM boosted DEN-induced liver damage [57]. OSM also enhanced the numbers of nodule formations, increased hepatic progenitor cell (HPC) markers, and decreased rat survival. Mechanistically, they demonstrated that OSM stimulated HPC activation by recruiting CD68^+^ macrophages to release TNF-α. In addition, OSM overexpression failed to promote HCC development in TNF-α^−/−^ rats, suggesting that OSM plays a key role in tumor progression, modulating the inflammatory microenvironment [57]. Accordingly, OSM was upregulated in liver specimens of cirrhotic patients, carrying or not HCC, with the highest OSM levels correlating with advanced tumor stages [58]. More recently, evidence for a key role of OSM has been provided for NAFLD-/NASH-related HCC, which is characterized by chronic inflammation. Interestingly, OSM was selectively overexpressed in hepatic cancer cells and progressively increased with the tumor grade, with serum OSM levels being enhanced as well and correlated with a poor outcome, suggesting OSM as a putative prognostic factor for NASH-associated HCC [59].

Regarding CCA, a recent study outlined the relevance of an inflammatory microenvironment in iCCA progression mediated in part by OSM, as described elsewhere in this review [60]. Co-cultures of tumor-associated neutrophils (TANs)/TAMs with iCCA cells promoted tumor progression through OSM and IL-11 production by TANs and TAMs, respectively. Moreover, a positive correlation between immune cell infiltration and p-STAT3 expression was proposed to represent a predictive parameter for poor prognosis in CCA patients [60]. Indeed, the literature reported for CCA tissues a strong correlation between OSM expression and tumor infiltration of immune cells, including M2 macrophages, as well as a positive correlation with immune checkpoints such as PD-L1 and CTLA-4 [61].

### 3.2. Role of OSM in Cell Proliferation and Tumor Growth

OSM was originally described as a cytokine able to inhibit the proliferation of cancer cell lines related to melanoma (A375 cells and SK-MEL-28), lung carcinoma (A549), neuroblastoma (HTB10), and embryonic lung (WI-26 and WI-38) [34,62], with a growth decrease frequently linked to stimulation of cellular differentiation. Moreover, OSM displayed its growth-inhibitory effects in breast cancer cell lines (MCF-7 and MDA-MB231) through activation of STAT1 and STAT3 transcription factors, as well as by modulating the mitogen-activated protein kinase kinase (MEK)/ERK pathways [63]. Accordingly, no tumor formation has been observed in mice transfected with OSM-secreting glioblastoma cells, suggesting an anti-tumorigenic effect of OSM [64]. An anti-proliferative role of OSM was reported in immune-deficient mice injected with human melanoma cells [65] and in a chondrosarcoma model, where local intra-tumor overexpression of OSM reduced tumor development and enhanced tumor cell apoptosis through the JAK3/STAT1 axis [66]. Regarding HCC, OSM increased apoptosis and decreased the clonogenicity and growth of SMMC-7721 [67] and HepG2 cells [59], by reducing the percentage of cells in the S phase due to an arrest at G0/G1. Moreover, xenografted mice injected with OSM-overexpressing HepG2 cells showed smaller tumors, but increased vascularization and spontaneous lung metastasis [59]. Similarly, in CD133^+^ HepG2 cells, OSM treatment resulted in an increased apoptosis, as identified through increased annexin V and cleaved caspase 3 levels [59].

On the other hand, data from the last decade also indicate OSM as a pro-carcinogenic cytokine. OSM has been associated with increased proliferation and tumor growth in endometrial [68], ovarian [31], prostate [69], and lung cancer [70]. In addition, elevated OSM levels were found in the serum and/or tumor mass and correlated with tumor progression, in brain [71] and colon cancers [72], myeloma [73], pancreatic cancer [74], and hepatoblastoma and HCC [58,59]. Accordingly, treatment with OSM of EpCAM^+^ HCC cells, but not CD133^+^ HepG2 cells, induced colony formation and cell proliferation [75,76]. OSM-supplemented media also led to an expansion rate of 3D organoids populated by adult-donor-derived hepatocytes or by HepG2 cells [77]. Moreover, OSM, through the activation of the STAT3, MEK, and JAK pathways, induced the expression of glucose-regulated protein 78 (Grp78) in HepG2 and Huh7 cells, a factor involved in proliferation, cancer progression, and the survival of tumor cells [78]. OSM could exert its biological effects in HepG2 also indirectly through the increase of membrane-associated and soluble IL-6R, although high concentrations of OSM may inhibit soluble IL-6R production [79], then possibly explain, at least in part, the opposite effects exerted by OSM in liver cancer. Finally, knockout mice carrying hepatocyte-specific gp130 deletion (gp130Δhepa) and subjected to DEN-induced acute liver injury showed a decrease of IL-6 and OSM levels, leading to a reduction in the inflammatory response, smaller tumors, and reduced tumor burden, suggesting a role for these gp130 cytokines in HCC progression. This effect was related to a switch from the STAT3- and transforming-growth-factor-beta (TGFβ)-dependent pathways to STAT5 activation, which impairs TGFβ-dependent mechanisms, thereby reducing HCC development [80].

Regarding CCA, an interesting study showed an increased proliferation and colony-forming ability of iCCA cells following exposure to conditioned medium (CM) collected from TANs and TAMs isolated from iCCA specimens, as well as after exposure of iCCA cells to CM derived from TAN-TAM co-cultures [60]. Remarkably, OSM and IL-11 were the most abundant cytokines produced by TANs and TAMs, respectively, and their levels were enhanced in TAN-TAM co-cultures. Accordingly, co-injection of iCCA cells with either TANs, TAMs, or TAN-TAM co-cultures in xenograft experiments resulted in larger tumors than xenografts consisting of only iCCA cells. The protumorigenic action of TANs and TAMs was dependent on the activation of STAT3 in iCCA cells [60]. It is then conceivable that OSM, produced by cells of the TME, could contribute to iCCA growth and progression, with TANs, TAMs, and p-STAT3 levels being independent predictors of patient prognosis.

### 3.3. OSM and Cancer Progression: Cancer Stem Cell Features, Epithelial–Mesenchymal Transition, and Angiogenesis

OSM has been reported to induce EMT [81], a crucial process that drives tumor metastasis [82], and CSCs’ plasticity program, with CSCs showing self-renewal, tumor initiation, and long-term repopulation potential properties [83]. OSM-dependent EMT and stem-like features require STAT3 activation [29,81]. OSM-activated STAT3 cooperates with TGF-β to induce mesenchymal stem cells’ properties in breast cancer [84] and in pancreatic cancer [81]. Along these lines, several studies demonstrated a critical role for OSM in metastasis in breast, lung, and gastric cancer [85,86,87].

Concerning HCC, overexpression of OSM in HCC cells or treatment of HCC cells with human recombinant OSM resulted in a rapid activation of STAT3 and induction of typical EMT changes, sprinkling from cell clusters, loss of cell-to-cell contacts, and acquisition of mesenchymal-like features. OSM inhibits E-cadherin and results in increased matrix metalloproteinases 2 (MMP2) and transglutaminase 2 (TGM2) protein levels, thus enhancing HCC cell invasiveness [59], similar to a study in which overexpression of OSMR in squamous cell carcinoma triggered TGM2/integrin–α5β1 interaction, leading to an enhanced aggressiveness [88]. In addition, xenograft experiments in nude mice showed the formation of spontaneous lung metastasis following injection with HepG2 cells overexpressing OSM; moreover, EMT markers were increased in cancer nodules and correlated with OSM expression in a DEN/choline-deficient L-amino acid (CDAA) hepatocarcinogenic protocol [59].

Remarkably, higher levels of OSMR are expressed in epithelial-cell-adhesion-molecule (EpCAM)-positive tumor cells displaying CSC properties. In EpCAM^+^ HCC cells, OSM induced a decrease of stemness markers, (EpCAM, albumin, and cytokeratin-19) and an increase in albumin expression by the activation of STAT3, indicating that this cytokine plays a role in hepatocyte differentiation. Moreover, in nude mice injected with EpCAM^+^ HCC cells, a strong inhibition of tumor growth was achieved when 5-fluorouracil (5-FU) and OSM were administered together. Indeed, OSM increased the chemosensitivity of these cells, boosting the antitumor action of 5-FU in HCC and inducing the shift of EpCAM^+^ CSCs into EpCAM^−^ non-CSCs, highly sensitive to 5-FU [76]. Similarly, OSM induced expression of CSCs’ differentiation-related markers in CD133^+^ HepG2 cells, also inhibiting cell invasion, effects enhanced by the combined treatment with salinomycin, a chemotherapeutic agent targeting CSCs [75]. Overall, OSM may display divergent effects in HCC, depending on the expression of CSC features of HCC cells.

In cholangiocarcinoma, as mentioned [60], TAN/TAM co-cultures produce high levels of OSM (TANs) and IL-11 (TAMs), which activate the STAT3 signaling pathway in iCCA cells, sustaining invasiveness. In addition, the co-injection of TANs, TAMs, and STAT3-knockout iCCA cells in xenograft experiments reduced metastasis, suggesting that STAT3 mediates the tumor-promoting effects of these cytokines in iCCA cell lines. However, another study performed on a limited number of patients (*n* = 30) reported a positive correlation between lymphatic and distant metastasis, vascular invasion, tumor stage, and low levels of OSM [61].

Regarding the promotion of angiogenesis, OSM is able to activate STAT3, and vascular-endothelial growth factor (VEGF) represents the main STAT3-regulated gene product [89]. Overexpression of OSMR in cervical SCC cells induced a proangiogenic phenotype [90], and in both osteosarcoma and endometrial cancer cells, OSM boosted angiogenesis via JAK/STAT3 pathway activation and enhanced expression of VEGF [68,91]. In relation to HCC, OSM, by activating the STAT3, ERK1/2, and p38 pathways, was able to induce HIF1α in normoxic conditions, enhancing the expression and release of VEGF and promoting angiogenesis in vivo and in vitro [59,92]. Moreover, in vivo experiments showed that tumors of mice injected with HCC cells overexpressing OSM are characterized by a more widespread vascularization [59].

The biological effects of OSM are resumed in Table 1.

## 4. Oncostatin M and Liver Stromal Cells

As mentioned above, OSM is mainly produced by cells of innate and adaptive immunity, such as activated T lymphocytes, monocytes, and macrophages. Additionally, OSM elicits several biological functions in different cell types, including non-parenchymal cells and cancer cells. Several literature data have reported a great impact of human liver stromal cells on chronic liver disease progression and tumor development, with a critical role for activated HSCs and resident macrophages (KCs). As previously reported [56], OSM protein is expressed in KCs, variably in normal liver, but ubiquitously and more abundant in cirrhosis.

In this scenario, Henkel and colleagues [94] revealed that the inflammatory mediator prostaglandin E2 (PGE2), usually released during inflammation associated with obesity and/or metabolic disorders, is able to indirectly affect liver damage by inducing OSM production by KCs. OSM, in turn, through paracrine signaling involving STAT3 phosphorylation and SOCS3 activation, may contribute and enhance insulin resistance and hepatic steatosis accompanying NASH.

Recently, Lu and collaborators [95] showed a novel role of KCs not associated with liver immunological surveillance. This study showed an involvement of KCs in initiating and prolonging liver regeneration by releasing OSM. In particular, OSM acts as a regeneration-promoting cytokine to initiate and maintain the progression phase of liver regeneration by inhibiting the TGF-β2/Smad pathways and, thus, sustaining hepatic proliferation.

In this connection, a study by Matsuda and colleagues [47] outlined a role of OSM in the progression of liver fibrosis through the regulation of the cooperation between liver-resident KCs and/or bone-marrow-derived macrophages (BMDMs) and HSCs/MFs. Specifically, they showed a direct role of OSM in partial activation of HSCs by inducing morphological changes of cells and overexpression of TIMP-1. Moreover, OSM seems to be able to activate HSCs indirectly through the release of TGF-β1 and PDGF-B by resident and/or recruited macrophages stimulated by OSM, which led to increased expression of collagen type I in HSCs. Accordingly, a study by Foglia and collaborators [96] showed corroborating data on the role of OSM in the regulation of pro-fibrogenic phenotypic responses (i.e., migration and wound healing) of HSCs, with a focus on its involvement in NAFLD/NASH progression. In particular, in this study, the authors investigated the expression of OSM and its receptor (OSMRβ) in three different models of experimental NASH, as well as in NASH patients.

Along these lines, the recruitment of HSCs and the infiltration of immune cells in the tumor microenvironment represent a risk factor for HCC development and progression because HSCs can promote and favor the switch of macrophages from the M1 anti-inflammatory phenotype to the M2 pro-tumorigenic and immunosuppressive signature [97]. In addition, HSCs help tumor cells escape immune surveillance by improving the recruitment of regulatory T cells (Tregs) and MDSC populations [97] 

Among non-parenchymal cells potentially contributing to chronic liver disease progression, it is necessary to consider liver sinusoidal endothelial cells (LSECs). A study by Arshad and colleagues [45] reported that OSM is able to increase the expression of IL-33 in liver endothelial cells derived from mice infected with adenovirus encoding OSM (AdOSM). Of relevance, ST2, the IL-33 receptor, was overexpressed in human CCA, and it has been demonstrated that IL-33, via IL-6, has a critical role in promoting CCA development in mice with a constitutive activation of Akt (myr-Akt) and Yap (YapS127A) [45,98,99].

Different from data concerning macrophages, the current literature on HSCs and LSECs only reports information about the ability of OSM to affect the behavior and the phenotypic responses of these stromal cells, but no data are available about their ability to produce and release OSM in the tumor microenvironment.

## 5. OSM and Therapeutic Strategies for HCC and CCA

Drug strategies to target OSM biological activities fall into four categories: (i) monoclonal antibodies that directly block the cytokine (GSK315234 [100] or GSK2330811 [101]); (ii) cytokine antagonist (Jorcyk et al., date of patent 24 January 2017, *Oncostatin M (OSM) antagonist for preventing cancer metastasis and IL-6 related disorders*, U.S. Patent USO09550828B2, https://patentimages.storage.googleapis.com/e7/55/a4/026371eca35619/US9550828.pdf); (iii) small molecules targeting LIFRβ (EC359 [102]); (iv) small molecules interfering with OSMR through gp130 and the JAK–STAT3 pathway. It was indicated that the use of AG490 (a JAK2-specific inhibitor) led to reduced tumor numbers and sizes due to STAT3 blocking in HCC rats [103]. Other JAK inhibitors (AZG1480 and TG101209) have suppressed HCC growth [104]. Moreover, blocking the phosphorylation/activation of STAT3 through small molecule inhibitors (C188-9, curcumin, OPB-31121, S31-201, LLL12 and AZD9150, some included in clinical trials) resulted in reduced proliferation of HCC cell lines, a reduced size of tumors in xenograft experiments, and the reduced apoptosis of HCC cells [104]. Moreover, OSM treatment increased the chemo-sensitivity of these cells to 5-FU in liver cancer cells with stemness-like features [76]. Additionally, OSM and salinomycin (Sal) combined treatment showed synergistic effects in suppressing proliferation and inducing apoptosis of CD133+ liver CSCs. Finally, OSM and STAT3 may represent potential therapeutic targets and prognostic markers for CCA [60,61]. For CCA treatment, several non-peptide SH2 domain or STAT3 inhibitors have been identified, including STA-21, IL-6, STTT, TIC, C188-9, OPB-31121/51602, WP1066, S3I-201, BP-1-102, STX-0119, and HJC0123 (see [105] and the references therein), although the efficacy of these agents requires further confirmation. The inhibitor AZD9150, which blocks the interaction of STAT3 with the promoter of related target genes, is expected to be efficient in CCA [105].

## 6. Conclusions

The clinical outcome of PLCs remains poor, with the majority of patients presenting an advanced stage not eligible for curative therapy. At present, early-stage tumors are approached with surgical therapies, whereas trans-arterial or systemic therapies are for intermediate and advanced stages, respectively. Single-agent therapies have not afforded satisfactory results, likely because of the complexity of the tumor microenvironment and heterogeneity of cancer cells. The use of molecular immuno-therapy in combination with systemic drugs represents the present challenge to treat advanced PLCs. According to current knowledge, OSM and related signaling pathways, known to affect tumor growth by modulating inflammatory responses, angiogenesis, proliferation, as well as acquisition of the metastatic phenotype, might represent novel critical targets for therapeutic strategies to treat PLC.

## Figures and Tables

**Figure 1 cancers-14-04211-f001:**
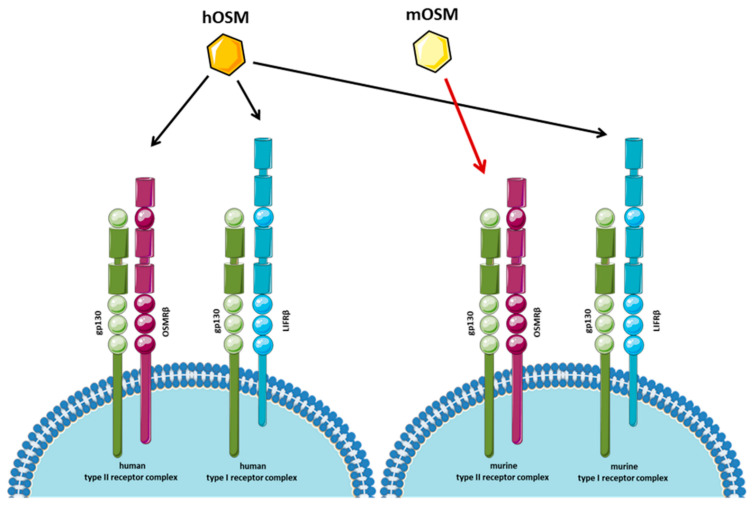
Different receptor binding for human and mouse OSM. OSM binds two different heterodimeric receptors: the type I receptor, formed by gp130 and LIFRβ, and the type II receptor, formed by gp130 and the OSM receptor β (OSMRβ). Human OSM can bind to both gp130/OSMRβ and gp130/LIFRβ, whereas murine OSM interacts only with the type II receptor. OSM: oncostatin M; LIFRβ: LIF receptor β.

**Figure 2 cancers-14-04211-f002:**
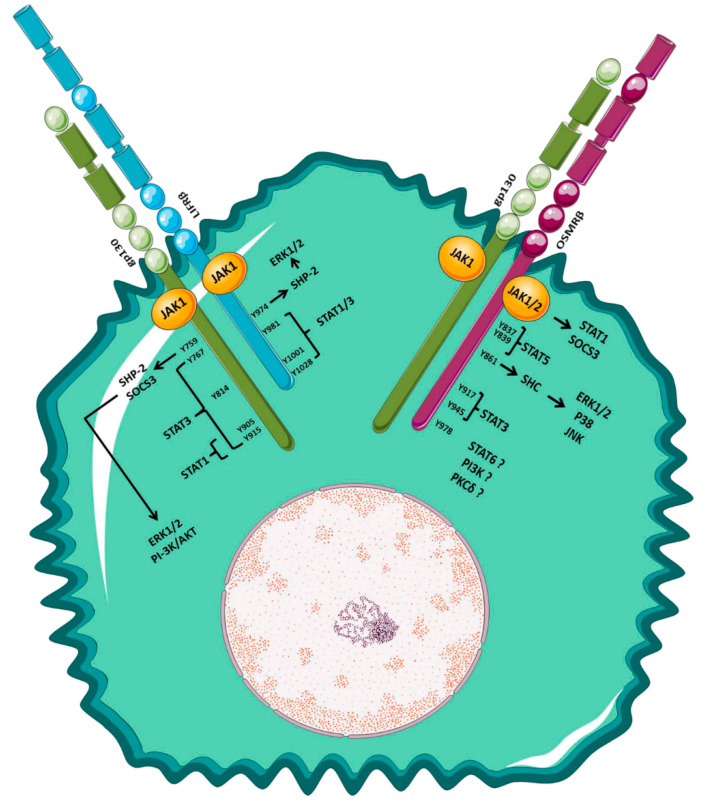
Overview of the OSM signaling pathways. OSM–receptor hetero-dimerization drives JAKs’ recruitment. The C-terminal region of receptor types I and II contains tyrosine residues, which, phosphorylated by JAK1/2, act as a docking site for STAT1 and STAT3. OSM can also activate other downstream proteins such as ERK1/2, p38, JNK, the PI3K/Akt pathway, and PKCδ. OSM: oncostatin M; JAKs: Janus kinases; Erk1/2: extracellular-regulated kinase ½; PI3K: phosphatidylinositol 3-kinase; STAT: signal transducer and activator of transcription; JNK: Jun N-terminal kinase; PKC: protein kinase C.

**Table 1 cancers-14-04211-t001:** Biological effects of oncostatin M in liver cancer.

Tumor Type	Inflammation	Cancer Cell Proliferation/Tumor Growth	CSCs’ Features and EMT	Angiogenesis
**HCC**	Induces macrophage recruitment and TNFα secretion [93]	Decreases proliferation and increases apoptosis in SMMC-7721 and CD133^+^ HepG2 cells [67]. Induces Grp78 expression in HepG2 and Huh-7 cells [78]. Induces cell proliferation in EpCAM^+^ HCC cells [75].	Induces a decrease of stemness markers in EpCAM^+^ HuH1 and HuH7 cells. Increases the chemosensitivity of EpCAM^+^ HCC cells in xenograft mice [76]. Induces the expression of CSCs’ differentiation-related markers in CD133^+^ HepG2 cells and inhibits cell invasion [75].	Induces HIF1 upregulation and VEGF gene overexpression. Xenograft mice injected with OSM-overexpressing HepG2 cells show a more extensive tumor vascularization [59].
**CCA**	Secreted by co-cultured TANs and TAMs [60].	Promotes proliferation in iCCA cells via STAT3 [60].	Induces iCCA cell invasion via STAT3. Xenograft mice co-injected with TANs, TAMs, and iCCA cells show less metastasis when STAT3 is knocked down in iCCA cells [60].	

CCA: cholangiocarcinoma; CSCs: cancer stem cells; EMT: epithelial–mesenchymal transition: EpCAM: epithelial cell adhesion molecule; HCC: hepatocellular carcinoma; iCCA: intrahepatic cholangiocarcinoma; OSM: oncostatin M; STAT3: signal transducer and activator of transcription 3; TAMs: tumor-associated macrophages; TANs: tumor-associated neutrophils; TNFα: tumor necrosis factor α; VEGF: vascular endothelial growth factor.

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
