# Peer review of "Oncostatin M: From Intracellular Signaling to Therapeutic Targets in Liver Cancer"

_cancers, 2022, doi:10.3390/cancers14174211_

Round 1

Reviewer 1 Report

In the review, authors explicitly summarized the current knowledge available about the role of oncostatin M on the several hepatic events. Cytokines or other secreted proteins are well known to be important for construction of microenvironment for development hepatic disease or liver cancer. Oncostatin M is the one of important factors for the inter cellular communications between hepatocyte and other stromal cells. It is difficult to cover such an extensive topic, but the review is well-written and will be useful for the general readers. I would like to comment just a few point which should be considered by authors. 

  1. The review is well written about the role of OSM. But, since OSM works as secreted signal protein, reader will be wondered that from which cell OSM was secreted. Most of disease and cancer in the liver are controlled by the cytokine signals from several nonparenchymal cells such as LSEC, stellate cells (HSC), Kupffer cells or other immune-related cells. For example, authors described that “OSM induce EMT” in page 8 line 302. Basically, EMT of liver cancer cell was controlled by cytokine signals from cancer microenvironment consisting cells such as activated HSC or Kupffer cells. But this review little mention about other stromal cells, though OSM is secreted protein for paracrine signal. it's better to focus on them.
  2. In page 6 line 219, authors described “Matsuda and colleagues reported that OSM promoted ... and type I collagen deposition by activated hepatic stellate cells (HSC), as well as by favoring M2 phenotypic switch of macrophages”. But readers may not understand the detail. OSM activate HSC and macrophages? How? OSM is secreted from which cells? To ensure the accuracy and completion of a paper, authors better to describe more detail about the citing research through out the manuscript.

Author Response

We would like to thank the reviewer for her/his comment. As suggested by the reviewer, we have addressed in the revised manuscript the points raised by adding a paragraph (new paragraph 4 entitle “Oncostatin M and liver stromal cells”), not dealt with in the original version of the paper given the extensive of the topic and the length limits provided by the journal.

Reviewer 2 Report

The author said that OSM, known to affect tumor growth by modulating inflammatory responses, angiogenesis, proliferation as well as acquisition of metastatic phenotype, might represent novel critical targets for therapeutic strategies to treat PLC. This review paper covered all pathway and reaction of OSM in cancer and can explain the mechanism of OSM in cell. this review paper was excellent and will be useful of research in OSM.    

Author Response

We would like to thank the reviewer for her/his overall positive evaluation of our manuscript

Reviewer 3 Report

Comments to the manuscript entitled "Oncostatin M: from intracellular signaling to therapeutic targets in liver cancer"

Some suggestions are listed below.

1. include the meaning of Src line 149 page 4

2. 5-FU line 326, is mentioned before including the meaning line 328

3. Include the meaning of TANs and TAM line 334, meaning is mentioned later in line 359, table 1 footer

4. Homogenize the way of citing line 362, Jorcy R et al., and on page 222 Yang and collaborators

Author Response

We would like to thank the reviewer for her/his comments. As suggested, we have addressed in the revised manuscript the points raised by the reviewer.

  1. Include the meaning of Src line 149 page 4.

Point 1: src meaning has been added.

  1. 5-FU line 326, is mentioned before ,,,,,

Point 2: The full form of 5-FU has been shifted to line 324.

  1. Include the meaning of TANs and TAM line 334, meaning is mentioned later…

Point 3: The full form of TANs is present in line 239 , whereas the full form of TAMs is written in line 97

  1. Homogenize the way of citing…

Point 4: The way of cyting authors has been homogenized.

.